# OpenReview forum: "ASIDE: Architectural Separation of Instructions and Data in Language Models"
_ICLR.cc/2025/Workshop/BuildingTrust — BuildingTrust_

### Official Review · Reviewer_yo2E · 2025-02-22
**Reviews from yo2E**

**Rating:** 9
**Confidence:** 4

**Review:**

The topic of this paper is aligned with the goal of this workshop: as it focus on the one of the challenges that may cause the prompt injection attacks, i.e., intrinsic separation between instructions and data.

The paper propose an "architectural" change to purposely separate instructions and data using different embeddings, and show the competitive results on prompt injection benchmarks without dedicated safety training. The idea seems to be very straight-forward and easy to scale-up (as the proposed method only requires a different size of embedding layer for instruction and an adapted tokenizer, which don't require to retrain the PLMs from scratch).

Several concerns:

(1) the paper relies on a strong assumption of perfect instruction-data classification, however, in real-world settings, a token could function as both instruction and data (e.g., translate hello to french, where it can be considered as a instruction but also cover the required data), I'm curious about how the paper's solution can be adapted to these cases or authors may hold different opinions.

(2) there might be limited exploration of the fine-tuning alternatives. The paper might consider more alternatives like parameter-efficient tuning (LoRA, adapters) or other training-free approaches, which could increase the feasibility of applying ASIDE to larger, more complex models.

---

### Official Review · Reviewer_TNNo · 2025-03-01
**ASIDE introduces a novel embedding-based instruction-data separation technique to mitigate prompt injection but lacks evaluation against diverse adversarial strategies and multi-turn attacks. While effective in improving security, it requires more diverse prompt injection benchmark evaluations to assess real-world robustness. Rating: Clear Accept (Top 50%).**

**Rating:** 8
**Confidence:** 3

**Review:**

### Summary

The ASIDE paper introduces an architectural modification for LLMs that enforces instruction-data separation to mitigate prompt injection attacks. It achieves this by using separate embeddings for instructions and data, with data embeddings rotated to prevent execution of injected commands. ASIDE significantly reduces attack success rates (ASR) while maintaining instruction-following performance and can be integrated into existing models with minimal fine-tuning.

### Strengths

ASIDE introduces a novel architectural modification that explicitly distinguishes executable instructions from user data, preventing adversarial prompt injections at the embedding level.
ASIDE can be applied post-hoc to pre-trained models with minimal fine-tuning, making it a practical and computationally efficient solution.
Despite improving security, ASIDE maintains strong instruction-following capabilities,

### Weaknesses

The success criteria for prompt injection attacks in ASIDE’s evaluation are overly simplistic, making it easier to defend against structured attacks while ignoring more diverse adversarial strategies. In TensorTrust, ASR is high if the model outputs "Access Granted" after adversarial manipulation; in Gandalf, ASR is high if the model leaks the password "PLANETARY"; and in Purple, ASR is high if the model outputs "purple", despite explicit instructions not to do so. While these benchmarks test basic prompt injection vulnerabilities, they lack attack diversity, as they only evaluate simple rule-breaking scenarios without considering adaptive adversarial techniques, multi-turn attacks, or stealthy manipulations. To improve robustness testing, more diverse benchmarks should be incorporated, such as Microsoft BIPIA, which evaluates adversarially optimized indirect prompt injections, WILDGUARD (AllenAI), which focuses on stealthy adversarial manipulations in real-world LLM applications, and HackPrompt, which tests jailbreak techniques and adversarial red teaming prompts. More advanced and diverse benchmarks are necessary to accurately measure ASIDE’s ability to resist prompt injection attacks in real-world scenarios.

Microsoft BIPIA https://github.com/microsoft/BIPIA
WILDGUARD https://huggingface.co/datasets/allenai/wildguardmix
Hackprompt https://huggingface.co/datasets/hackaprompt/hackaprompt-dataset.

How well does ASIDE handle multi-turn prompt injections, long context and nested instructions?

Not clear if a different rotation would be more or less effective. Also, Is there any other  best way to achieve separation apart from rotation?

Furthermore, ASIDE can be integrated into already existing language models with minor overhead. There is no concrete discussion about overhead.



### Clarifications that did not affect score

Why is high temperature of 0.7 used for evals?

---

### Official Review · Reviewer_rFci · 2025-03-01
**Assessment of ASIDE: A Framework for Instruction-Data Separation in LMs with Mixed Results on Prompt Injection Benchmarks**

**Rating:** 6
**Confidence:** 2

**Review:**

This paper introduces a framework - ASIDE - for working with LMs to help separate instructions from data. The idea is to use 2 separate token embeddings – (1) executable instructions and (2) for tokens in non-executable data. The embedding is initialised as a rotated version of the instruction embedding, which helps the model learn to process instructions and data differently. This is tested through evals on instruction-data separated metrics and prompt injection benchmarks.

Questions
- How does the computational cost of ASIDE compare to standard models? Does the double embedding size significantly impact inference speed or memory requirements?
- Have you explored different rotation angles beyond 90 degrees? Is there an optimal angle for instruction-data separation?
- How does ASIDE perform when combined with specialized safety training or adversarial examples? Could this further improve robustness?
- How does the model behave with more complex hierarchies beyond the binary instruction/data distinction? Could this approach be extended to handle multiple privilege levels?
- Have you tested ASIDE on other model architectures beyond Llama, such as Mistral or other transformer variants?

Doubts
- Not sure if their training approach is comprehensive enough - they only used standard Alpaca data
- The results on some prompt injection tests were mixed (like in Table 2 for Completion attacks)
- Didn’t see much comparison with other security approaches
- There's no analysis of how much extra computation this requires
- I wonder if this approach would work on non-Llama architectures

---

### Decision · Program_Chairs · 2025-03-04

Accept